# Advances in Glioblastoma Therapy: An Update on Current Approaches

**DOI:** 10.3390/brainsci13111536

**Published:** 2023-10-31

**Authors:** Ramcharan Singh Angom, Naga Malleswara Rao Nakka, Santanu Bhattacharya

**Affiliations:** 1Department of Biochemistry and Molecular Biology, Mayo Clinic College of Medicine and Science, 4500 San Pablo Road South, Jacksonville, FL 32224, USA; angom.ramcharan@mayo.edu (R.S.A.); nakka.naga@mayo.edu (N.M.R.N.); 2Department of Physiology and Biomedical Engineering, Mayo Clinic College of Medicine and Science, 4500 San Pablo Road South, Jacksonville, FL 32224, USA

**Keywords:** glioblastoma multiforme, therapy, radiation, surgery, nanocarriers, immunotherapy, vaccines

## Abstract

Glioblastoma multiforme (GBM) is a primary malignant brain tumor characterized by a high grade of malignancy and an extremely unfavorable prognosis. The current efficacy of established treatments for GBM is insufficient, necessitating the prompt development of novel therapeutic approaches. The progress made in the fundamental scientific understanding of GBM is swiftly translated into more advanced stages of therapeutic studies. Despite extensive efforts to identify new therapeutic approaches, GBM exhibits a high mortality rate. The current efficacy of treatments for GBM patients is insufficient due to factors such as tumor heterogeneity, the blood–brain barrier, glioma stem cells, drug efflux pumps, and DNA damage repair mechanisms. Considering this, pharmacological cocktail therapy has demonstrated a growing efficacy in addressing these challenges. Towards this, various forms of immunotherapy, including the immune checkpoint blockade, chimeric antigen receptor T (CAR T) cell therapy, oncolytic virotherapy, and vaccine therapy have emerged as potential strategies for enhancing the prognosis of GBM. Current investigations are focused on exploring combination therapies to mitigate undesirable side effects and enhance immune responses against tumors. Furthermore, clinical trials are underway to evaluate the efficacy of several strategies to circumvent the blood–brain barrier (BBB) to achieve targeted delivery in patients suffering from recurrent GBM. In this review, we have described the biological and molecular targets for GBM therapy, pharmacologic therapy status, prominent resistance mechanisms, and new treatment approaches. We also discuss these promising therapeutic approaches to assess prospective innovative therapeutic agents and evaluated the present state of preclinical and clinical studies in GBM treatment. Overall, this review attempts to provide comprehensive information on the current status of GBM therapy.

## 1. Introduction

There is a dearth of efficacious treatment modalities for persons diagnosed with GBM [1,2]. GBM, the most observed malignant CNS tumor, displays a miserable prognosis with a median survival of 12–16 months, even when subjected to multimodal treatment tactics encompassing radiation therapy, temozolomide administration, and maximal safe surgical reection [3] Over the past few decades, limited progress has been observed in high-grade glioma’s overall survival (OS) rates (classified as grades III and IV). The presence of notable inter- and intratumoral heterogeneity, microscopic invasion, and challenges distinguishing tumor boundaries from healthy brain tissue during surgical procedures are all pertinent issues associated with managing the GBM [3,4,5].

The foundation of glioma treatment depends on the tissue diagnosis, which is further boosted by molecular analysis to complement the histopathologic examination. The attainment of gross total resection remains the preferred outcome, where feasible. The degree of tumor resection at the onset plays a pivotal role in treating disease and OS outcomes for patients with high-grade glioma, surpassing all other currently accessible therapies [6]. The extent of resection (EOR) refers to the amount of contrast-enhancing tumor surgically excised and quantified using MRI postoperatively. Currently, the most widely accepted approach for treating GBM is a combination of temozolomide (TMZ), irradiation, and aggressive surgical resection of the tumor [7]. Given the challenges associated with early diagnosis and the highly invasive nature of tumor cells, achieving complete surgical excision poses a significant challenge [7]. Despite ongoing advancements in surgical imaging techniques, which have led to more extensive surgical resections, it is necessary to find a balance between aggressive removal of tumor tissue and preserving brain function, all while ensuring patients’ well-being and quality of life [2]. Including targeted irradiation leads to a median survival duration of 12.1 months [7]. The association between radiation exposure and significant systemic harmful consequences, including DNA damage and cognitive impairment, has been established [8]. The U.S. Food and Drug Administration (FDA) approved using bevacizumab, an anti-angiogenic drug, to treat recurrent GBM that has shown progression after prior therapy [9]. The outcomes of phase III trials demonstrated that incorporating bevacizumab into concurrent chemo-radiotherapy for individuals with newly diagnosed GBM led to a notable enhancement in progression-free survival (PFS). However, it did not significantly improve overall survival (OS) outcomes [9,10,11].

In 2015, the FDA approved the use of Optune^®^, a medical device that produces low-intensity and alternating tumor-treating fields. This approval was explicitly for treating newly diagnosed GBM when combined with TMZ [3]. Despite the implementation of this combined therapy, a significant proportion of patients continue to encounter tumor recurrence within 1 to 2 years following their first diagnosis. Regrettably, the available data does not substantiate the assertion that these treatments can extend survival durations. Hence, the ongoing struggle against GBM remains unresolved about the unfulfilled medical needs of individuals affected. 

Combinatorial approaches are increasingly being used to treat patients with multiple metastases. In conjunction with surgery, stereotactic radiosurgery and brachytherapy can be utilized to enhance local control for brain metastases [3]. Contemporary surgical techniques have facilitated the implementation of minimally invasive procedures, as shown by using laser interstitial thermal therapy, endoscopes, and tubular retractor devices. The molecular characterization of malignancies has facilitated the exponential expansion of targeted medications and immunotherapies. Understanding the impact of genetic subclassification on prognosis and treatment response has necessitated a shift in managing many tumors, including high-grade glioma. However, advancements in this area have not been impressive [12]. On the contrary, there has been a significant increase in available therapeutic options for solid tumor brain metastases, resulting in notable enhancements in patient survival rates and prognosis [3,13]. There is a need for a novel approach to address previously treated lesions, as the observed improvements in survival rates have been concomitant with a rise in the population of patients who exhibited satisfactory systemic control and functional status upon recurrence or advancement of the condition [3]. An emerging area of tumor therapy is immunotherapy [14]. Different immune checkpoint inhibitors have shown promising results in a number of malignancies. Both in pre-clinical and clinical trials, the traditional checkpoints PD-1/PD-L1, CTLA4, TIM3, and others have made impressive strides. Combination therapy is suggested due to the tumor microenvironment’s intricacy and the immune response’s modulation, particularly when anti-PD-1 and anti-CTLA4 therapy together show promising efficacy on recurrent GBM (rGBM). Traditional anti-PD-1 may be coupled with a number of targets, including TIM3 and BTLA.

Combining immunotherapy with another targetable mechanism has a similar attraction. The appeal of anti-CD276 studies in combination with bevacizumab stems from the documented association of CD276/B7-H3 with angiogenesis.

The extracellular domain of EGFR, which is only present in the EGFRvIII mutation, makes it the perfect particular antigen for both vaccines and CAR-T treatment. Given that single-target therapy causes recurrence and later resistance to the initial treatment, as was previously discussed. 

Recent research has provided evidence that extracellular vehicles (EVs) possess advantageous characteristics such as excellent biocompatibility, a high capacity for carrying drugs, extended circulation duration, efficient crossing of the blood–brain barrier (BBB), targeted delivery to affected areas, and effective transport of diverse cargos for the treatment of GBM [15]. Significantly, EVs possess physiological and pathological components derived from the originating cells, making them valuable biomarkers for molecularly monitoring the malignant advancement of GBMs [16]. Moreover, the utilization of nanocarriers presents the potential to enhance the effectiveness and safety of diverse therapeutic interventions through the synergistic combination of these treatments [17]. Several studies have addressed various critical aspects related to treating GBM, including the primary challenges associated with GBM treatment, the advantages of utilizing drug combinations with or without nanocarriers, the notion of improved permeability and retention effect in tumor targeting using nanomedicine, as well as the potential of nanodiagnostics and nanotherapeutics in GBM management [17,18]. Here, we review in detail the justifications for the prospective therapy’s use in treating GBM, discuss potential therapeutic targets, analyze the pre-clinical and clinical study’s current state, and review new therapies’ difficulties and probable futures.

## 2. Therapeutic Resistance in GBM

The poor prognosis in GBM is due to unique treatment limitations, including the intertumor and intratumor heterogeneity, which facilitates the selection of resistant subpopulations, the fortified location of the tumor, which hinders the delivery of therapeutics, as well as the induction of a strong local immunosuppression. The DNA repair enzyme O6-methylguanine-DNA methyltransferase (MGMT) can repair the alkylation damage of TMZ by eliminating O6-methylguanine adducts. As a result, the MGMT gene promoter’s methylation state has significant clinical implications [19,20]. Several known mechanisms underlie the therapy resistance in GBM. As determined by immunohistochemical staining, the presence of methylated MGMT promoters and subsequently lost MGMT protein expression responded more favorably to TMZ treatment [21]. Overall survival and progression-free survival (PFS) are both markedly increased by MGMT inactivation or silence [22,23,24]. MGMT is one of the most important prognostic indicators due to TMZ’s involvement in CCRT. Therefore, CpG dinucleotide methylation may prevent these transcription factors from binding and shorten the MGMT gene’s transcriptional activation time [25,26,27]. When activated in the hypoxic glioma initiating cells (GICs) niche, other transcription factors, such as hypoxia-inducible factor-1 (HIF-1), would increase MGMT expression [28,29]. Another important factor is the ATP binding cassette (ABC) transporter. There are 49 ABC transporter family members, divided into seven gene subfamilies called ABCA-G. Among the pumps found in tumor stem-like cells, ABCB1, Multidrug Resistance 1 (MDR1), ABCC1, Multidrug Resistance Protein 1 (MRP1), and ABCG2, Breast cancer-resistance protein 1 (BCRP1) are the most well-known. ABCG2 was first discovered and linked to subpopulations of multidrug-resistant, stem-like cells [30]. Since blocking ABCC1 improves therapeutic response in GBMs [31,32], ABCC1 is thought to be another factor in GBM recurrence. The tumor’s hypoxic niches are enriched with GICs [33]. Chemoresistance is brought on by low oxygen levels, which also increase the expression of MGMT, ABCC1, and ABCB1 [34]. The GBM cells became more susceptible to TMZ treatment when ABCB5 was knocked down [35]. The role of ABC transporters in therapy-resistant GICs has been linked to a number of pathways such as Sonic Hedgehog (SHH), Wingless Integrated (Wnt)-catenin pathway, Bcl-2, Akt, survivin, etc. [36], which implies that the ABC transporter is yet another target for which new treatments may be created [37]. The failure of anti-glioma drugs to cross the BBB is due to the protective boundary between the circulatory system and the extracellular space of the central nervous system which is mainly composed of endothelial cells that form a tight barrier along the wall of blood vessels and selectively limit the compounds that can cross into the parenchyma. Further, immune checkpoint inhibitors cannot work on these tumors because there is no pre-existing T cell infiltration. With novel, innovative combination treatment regimens being evaluated in preclinical and clinical trials, a combination therapy approach is being developed against GBM. Combination therapy and pharmacological synergism show potential for targeting heterogenous tumor. To maximize each treatment method’s anticancer potential, future research should concentrate on identifying synergistic interactions between chemotherapy, radiation, and immunotherapy. A subpopulation of GBM cells with stem cell-like characteristics (self-renewal, multi-lineage differentiation) can recreate the original tumor. In the perivascular niche, where they can be very resistant to chemotherapy and radiation [38,39,40], these glioma stem cells (GSCs) are seen. A favorable association between the density of GSCs and tumor-associated microphases (TAM), which suggests that GSCs may attract TAMs more effectively than their more differentiated neoplastic counterparts [41], further emphasizes the significance of GSCs to microglia recruitment. It has been shown that the tumor microenvironment’s growth-regulating signals are necessary for glioma proliferation [42]. After glioma radiotherapy, the elevation of Hypoxia-inducible factor-1 HIF-1 causes the recruitment of myeloid cells from the bone marrow. This is partly because chemokine stromal derived factor-1 SDF-1 and its receptor, C-X-C chemokine receptor type 4 (CXCR4), are activated. As a result, SDF-1 and CXCR4 activation encourages tumor recurrence and vasculogenesis. These results support the hypothesis that radiation plus AMD3100, a clinically licensed small molecule inhibitor of CXCR4 signaling, may be used to improve outcomes for glioblastoma [43].

## 3. Currently Used Therapeutic Approaches

The current approaches include surgery, chemotherapy, and radiation therapy in conjunction with treating brain tumors (Table 1). Recent improvements in brain tumor patient treatment plans have led to better outcomes for these patients.

## 4. Surgical Method of Removing the Tumor

Surgical resection has conventionally been seen as an essential element in treating brain tumors. The evidence so far suggests an associative correlation between maximizing surgical resection and increased life expectancy in patients diagnosed with low- or high-grade gliomas. However, it is essential to note that data are scarce about the specific influence of the extent of resection on patient survival [44]. Moreover, surgical resection is paramount in managing metastatic brain lesions, extending beyond their mere diagnostic value. According to a study published in 2013, a randomized experiment was conducted to compare the efficacy of radiation therapy alone with surgical excision of single brain metastases. The study’s findings revealed that the group receiving surgical treatment had improved life expectancy and quality of life outcomes [44].

## 5. Resection with Fluorescence Guidance

The fluorescence-guided approach leads to an enhanced extent of resection [45]. Multiple trials have demonstrated that 5-ALA effectively enhances the EOR in high-grade gliomas. Previously, Gandhi et al. published a systemic review and described that the observed rate of gross total resection (GTR) was found to be 76.8% (95% confidence interval, CI, 69.1–82.9%) when utilizing 5-ALA as a guide for surgical resection in patients with high-grade gliomas [46]. Using 5-ALA during surgical procedures resulted in a higher rate of GTR than conventional surgical methods. In addition, the administration of 5-ALA resulted in an extension of OS by three months and PFS by one month, as reported previously [46]. The study conducted by Golub et al. demonstrated that 5-ALA-guided resection exhibited superiority over conventional neuronavigation alone, as indicated by an odds ratio of 2.866 (95% CI: 2.127–3.863; *p* < 0.001) [47]. Another study by Haider et al. verified that using 5-ALA-guided resection effectively enhanced the EOR when tumors were found to be fully removable before surgery [48]. This technique resulted in a notable increase in EOR, with some cases achieving a complete removal rate of 100%. Furthermore, the integration of 5-ALA with the excision of lesions in eloquent structures resulted in a significant increase in the EOR, from 57.6% to 71.2%, compared to the use of iMRI alone [48]. Nevertheless, using different intraoperative adjuncts during the surgical excision of high-grade gliomas can potentially lead to improved EOR and extended OS and PFS.

## 6. Operative Resection in GBM

The management of GBM has experienced a significant paradigm shift due to the high incidence of isolated brain metastases in individuals with non-small-cell lung cancer (NSCLC) who acquire targetable mutations [49]. The effective management of metastatic disease necessitates the implementation of highly invasive surgical techniques for the treatment of brain metastases. Concerning mortality, adverse events, and quality of life, surgical resection, and stereotactic radiosurgery (SRS) have demonstrated comparable levels of safety and efficacy in treating single metastases. To evaluate the suitability of surgical management and adjuvant SRS, Hatiboglu et al. conducted a study and examined the patients who possess a high-performance score (Karnofsky Performance Status Scale (KPS > 70) [50]. SRS emerges as the principal therapeutic approach when surgical intervention is contraindicated for a tumor [51]. The necessity of surgical excision should be evaluated for accessible lesions in cases where many metastases are present. Patients who may be considered for surgical resection followed by SRS to the resection cavity include those with lesions larger than 3 cm, lesions that cause neurological deficits, lesions that result in significant radiographic mass effect, and lesions that cause impending interruption of cerebrospinal fluid (CSF) flow [52].

## 7. Laser Interstitial Thermal Therapy in GBM

Here a heat-delivering probe is used in laser interstitial thermal treatment (LITT), directed by MRI thermometry [53]. Heating tumor tissue causes targeted hyperthermic damage at the subcellular level, which results in coagulative necrosis [53]. The temperatures drop down exponentially beyond the treatment radius supported by the two approved LITT systems, NeuroBlate^®^ (Monteris Medical, Plymouth, MN, USA) and Visualase^®^ (Medtronic, Minneapolis, MN, USA) [54]. Real-time MRI thermometry monitors ablation temperatures simultaneously, and if temperatures rise beyond a preset threshold, treatment is immediately stopped [55]. Digital stereotactic navigation tools are used to map ablation trajectories, and effectiveness has been shown in lesions that are resistant to treatment or that are deep and difficult to reach through surgery. For treating patients with various intracranial malignancies, LITT has proven to be a secure and efficient method [56].

Since LITT is equally helpful for radiation-related alterations (such as radiation necrosis) and recurrent malignancies, it is particularly promising for recurrent brain metastases [54]. Additionally, it enables tissue samples to assist in differentiating radiation necrosis from recurrence, assisting in the direction of subsequent postoperative therapy decisions. With a median PFS of 37 weeks, Rao et al. successfully applied an LITT paradigm to target recurring metastatic lesions under 5 cm [57]. With reported local tumor control rates as high as 77.4%, LITT has met the requirements for local control in managing treatment-resistant metastases [55]. When at least 80% of the tumor is eliminated, according to multicenter research looking at the use of LITT on recurrent brain metastases that had previously undergone SRS, there is no disease progression [58]. 

In addition, LITT can be used to treat high-grade gliomas that cannot be surgically removed. Ivan et al.’s meta-analysis showed that an average % percentage of tumor ablation volume of 82.9% resulted in OS and PFS of 14.2 and 5.1 months, respectively [59]. Ablation volume and resection extent have been generally equated, and they can be thought of similarly in terms of efficacy. Off-target damage is reduced by the stereotactic placement of the probe with image guidance and real-time thermometry. LITT can also be used for isolated metastases resistant to radiosurgery or surgery. By rupturing the blood–brain barrier, LITT may also improve adjuvant treatments such as systemic chemotherapy [55,60]. Several academic institutes are rapidly adopting LITT into their treatment paradigms despite insufficient practice standards and evidence-based support for LITT [52].

## 8. Chemotherapy in GBM

The following section overviews the most frequently used chemotherapeutic drugs for treating primary brain tumors. The underlying tumor type and the presence of systemic illness influence the choice of chemotherapy strategy for metastatic brain cancers. It is outside the purview of this article to explore each of the many agents in detail. A comprehensive list of FDA-approved drugs used in treating brain tumors has been listed in Table 2.

## 9. Temozolomide

A DNA alkylating drug called temozolomide (TMZ) results in poor DNA repair and cell death. TMZ is an oral medication that can be given in a variety of clinical settings, including inpatient, outpatient, or at home [61]. The side effect profile for TMZ is largely positive. The concurrent treatment of TMZ with radiation has been shown to be safe in a 2002 phase II study of individuals with newly diagnosed GBM [62]. After radiotherapy, adjuvant monotherapy was given for an additional six cycles, indicating enhanced survival. Confirmatory research conducted in 2005 found that radiation therapy combined with TMZ resulted in a 2-year survival rate of 26.5%, with a median survival of 14.6 months, as opposed to radiation therapy alone, which had a 2-year survival rate of 10.4%, with a median survival of 12.1 months. The “Stupp protocol” is the name of the treatment plan, which is widely used in patients with GBM. Because these changes restricted the repair of TMZ-induced DNA damage, patients with methylation MGMT mutations benefited more from TMZ treatment [63]. Given the extensive use of TMZ and the GBM’s very diverse and mutagenic nature, it is quite typical for these deadly tumors to acquire TMZ resistance. Sadly, more than 50% of GBM patients receiving TMZ do not benefit from the treatment, and there are few other predictors of TMZ response than MGMT status [64,65]. The fact that TMZ resistance can either be an intrinsic trait of some cancers or develop after initial treatment makes it more difficult to understand and treat [66]. The treatment of GBM is still severely hampered by TMZ resistance, which also worsens the prognosis. The complicated interplay of various molecular pathways that contribute to the development of TMZ resistance has been described by Singh et al. [67]. To overcome TMZ resistance and eventually enhance patient outcomes, it is expected that a deeper comprehension of the dysregulated pathways utilized by GBM cells will lead to the development of innovative and more potent therapeutic methods. 

## 10. Vincristine, Procarbazine, and Lomustine

Combination therapy with procarbazine, lomustine, and vincristine is recommended in gliomas [68,69]. A vinca alkaloid called vincristine interferes with mitosis and the production of microtubules [70]. Patients with anaplastic oligodendroglioma and anaplastic oligoastrocytoma who received vincristine in addition to radiation therapy had longer PFS than those who received radiation therapy alone, according to a phase III study with a 3-year follow-up [71]. After a long-term median follow-up of 140 months, combination therapy showed significantly longer OS and PFS than patients who received adjuvant radiation first [68,72,73]. The alkylating drug procarbazine individually showed fatigue, anorexia, and myelosuppression as the side effects.

Similarly, some side effects of the nitrosourea alkylating drug lomustine are myelosuppression, nausea, tiredness, and pulmonary fibrosis. Combined radiation treatment with PCV significantly improved PFS compared with radiation and Temozolomide for IDH-mutant AA. However, radiation and TMZ were found to be better tolerated.

## 11. Bevacizumab

Bevacizumab is a monoclonal antibody to vascular endothelial growth factor and is known to prevent the formation of tumors [74]. It is typically given intravenously once every two weeks. Both a monotherapy and a combination with other adjuvant medicines are options for its administration. There are several possible side effects from bevacizumab that have been registered, including bleeding, GI perforation, slowed wound healing, and intensifying chemotherapy’s cytotoxic effects. However, a phase II trial showed improved radiographic response and median OS of 31 weeks with bevacizumab monotherapy in patients with recurrent GBM [75]. A recent review by Fu et al. has suggested that combining bevacizumab with novel treatments like tumor-treating field (TTF) and administration at first recurrence may optimize the therapeutic efficacy in GBM [74]. The OS effects of bevacizumab + radiation vs. bevacizumab monotherapy have been compared in different studies [76]. These retrospective studies claimed that radiation + bevacizumab improved the rGBM prognosis by boosting OS [77,78,79].

Bevacizumab plus re-surgery improved OS, according to a retrospective study by Yamaguchi et al. in 2021 (mOS, Cytoreductive surgery + bevacizumab vs. bevacizumab, 16.3 months vs. 7.4 months, *p* = 0.0008) [80], but a subsequent retrospective study in 2017 found no differences between bevacizumab combination and single regimen groups [81].

The use of bevacizumab for treating patients with recurrent GBM has been rejected by the European Medicines Agency (EMA), with one of the reasons being a lack of positive benefit–risk for bevacizumab [82].

## 12. Carmustine

A nitrosourea DNA alkylating chemical called carmustine blocks DNA replication and transcription—the intravenous administration of carmustine. Myelosuppression, weariness, nausea, vomiting, danger of pulmonary damage, and other side effects make carmustine much less attractive than temozolomide. With the addition of carmustine or other comparable nitrosoureas, surgery and radiation therapy be beneficial by meta-analysis [83]. However, carmustine was incorporated into a biodegradable polymer wafer and inserted into the surgical cavity at the time of surgical tumor removal to improve efficacy and reduce adverse effects. A second investigation confirmed similar findings [84]. In 1995, a placebo-controlled trial for recurrent gliomas showed a median survival of 31 weeks using the biodegradable polymer wafer against 23 weeks with placebo [85]. Carmustine-based chemotherapy leads to common side effects of such as nausea/vomiting and hematotoxicity. Pulmonary fibrosis is the most dreaded side effect, leaving carmustine preference over other cytotoxic drugs controversial [86]. 

## 13. Methotrexate

Dihydrofolate reductase is irreversibly bound and inhibited by methotrexate (MTX), which prevents DNA synthesis, repair, and cellular replication. MTX can be administered orally, intravenously, or intrathecally for oncologic therapy. A potentially effective treatment for patients with GBM is intrathecal MTX administration mixed with systemic chemotherapy. This, treatment has mild treatment-related side effects [87]. Numerous single-use intrathecal chemotherapeutics, including topotecan, MTX, and cytarabine, have been the subject of completed clinical trials that examined their potential applications. Despite the positive safety assessment, none of the single-use medications have been demonstrated to significantly increase the survival rate of Leptomeningeal disease (LMD) patients [88]. Most intrathecal drugs are single-use for LMD patients [89] reported that concurrent intrathecal MTX and liposomal cytarabine for solid tumors that developed LMD showed a median non-GBM OS of 30.2 wk, thereby demonstrating a possible strategy of multidrug intrathecal chemotherapy. A study demonstrated that MTX shows an extensive up-regulation of CD73 expression and immunosuppressive capability in GBM tumor tissue [90].

## 14. Limitations

As summarized in Wu et al.’s [2] review, hematologic toxicity is the most frequent negative side effect of TMZ. A range of 10% to 20% of patients have been found to have thrombocytopenia [91]. A thrombopoietin receptor agonist, Romiplostin, was added to adjuvant concurrent chemoradiation therapy (CCRT) in a Phase II clinical trial, and this enhanced the rate of successfully completed regimens [92]. There are also less frequent nonhematologic toxicities with TMZ, including nausea, anorexia, tiredness, and hepatotoxicity [93].

## 15. Radiation Treatment in GBM

The utilization of whole-brain radiation therapy (WBRT) is on the rise for individuals who are not eligible for surgical or Stereotactic radiosurgery (SRS) interventions since it has been shown to significantly enhance the survival rates of patients with brain metastases. Although WBRT has been found to be successful in attaining control over intracranial conditions, some studies have indicated a correlation between WBRT and a loss in neurocognitive function. This decline has been shown to impact the long-term cognitive status and overall quality of life of individuals undergoing WBRT [94,95]. Over the past decade, there has been a notable movement in the approach to palliative care for oncologic patients, with a greater focus on long-term oncologic benefits. This trend has been driven by therapeutic breakthroughs, resulting in a considerable departure from the use of WBRT and a move towards high-dose targeted radiation to enhance tumor control rates. Additionally, this approach is being explored as a cost-effective substitute for SRS.

Like the surgical resection approach, systemic medication is employed in conjunction with SRS to attain localized CNS control and impede the dissemination of metastatic disease. According to a recent study [96], it is recommended to employ SRS for the management of smaller lesions measuring 3 cm or less, particularly those with low edema or located in surgically challenging regions. In a recent phase 3 experiment conducted across many centers, a comparison was made between the cognitive outcomes and survival rates of patients who underwent SRS and those who received WBRT. In a retrospective study conducted in 2019, Nguyen et al. proposed the utilization of a single-fractioned partitioned SRS technique employing the Gamma Knife Icon Spatially Partitioned Adaptive Radiosurgery (GKI-SPARE) [97,98]. This technique offered a dosimetric benefit compared to the use of HA-WBRT for treating 10–30 metastases. The recommended adjuvant therapy of choice is SRS, and individuals who are not eligible for SRS or who have leptomeningeal spread are required to undergo WBRT. 

Recent studies have indicated no significant disparities in local control outcomes when comparing surgical resection and SRS as standalone treatment modalities [50,98]. Nevertheless, the highest level of local control is attained with the combination of radiosurgery and surgical excision. Therefore, the paradigms above serve as guiding principles for determining the initial treatment approach for patients diagnosed with brain metastases [94]. Enhanced performance ratings substantiate the efficacy of this integrated therapeutic approach, enhanced local control, and reduced reliance on steroids [94,99]. 

The current body of literature supports the utilization of several fractions to enhance the treatment effectiveness for brain metastases despite the feasibility of deploying either single-fraction or three-fraction SRS. As a result of these considerations, our standard protocol entails administering a total dose of 27 Gy in three fractions to metastatic brain sites, excluding lesions located in the brainstem and tumors exceeding 3 cm in size or exhibiting significant vasogenic edema. According to a published study, implementing a 2 mm margin surrounding the resection cavity in SRS significantly decreases local failure rates from 16% to 3% for 12 months [100].

Individuals who would typically not be considered suitable candidates for SRS due to large tumor sizes or lesions located in the eloquent cortex can experience advantages from using hypofractionated stereotactic radiosurgery (HF-SRS). It has emerged as a viable therapeutic option for patients requiring high radiation doses, with the added benefit of reducing negative neurocognitive effects. This is achieved by administering a sequence of low-dose treatments, resulting in a higher cumulative radiation dosage. According to a study conducted in 2019, administering a dose of 30 Gy for these lesions is generally considered safe and beneficial, divided over 5–6 sections [101]. This treatment approach helps minimize the occurrence of radiation necrosis while ensuring satisfactory tumor control.

According to recent research, there is evidence suggesting that radiation therapy possesses immunomodulatory properties, which can alter the microenvironment of tumors and consequently lead to the reactivation of the immune system [102]. Ongoing research endeavors are being undertaken to ascertain the impact of immunomodulation on the efficacy of immunotherapy, alongside investigating the consequences of surgical resection and radiation therapy.

## 16. Gamma Knife

The Gamma Knife (GK) utilized a stereotactic radiotherapy technique that enhances radiation delivery precision for localized malignancies, whether primary or metastatic. A multiheaded cobalt unit is used to distribute GK. One benefit of GK is that it just requires one treatment session and has fewer radiation effects on nearby healthy tissue. Given the focused beam field, greater accuracy is required for immobilization; patient immobilization is accomplished by using external frames (with an accuracy of 1–2 mm) or custom-fit masks (with an accuracy of 2–3 mm) [103]. GK-mediated stereotactic radiotherapy, combined with bevacizumab, has safely treated focal GBM recurrence [77]. The study also shows a lack of radiation injuries and improvements in both PFS and OS. Due to the retrospective character of our study and the possibility of selection bias, these observations are encouraging and call for further research. Future research may also help stratify the risk factors for radiation injury in patients who have had radiosurgery, explain the benefits of bevacizumab when combined with SRS, and shed light on utilizing targeted therapy in this patient population [77]. Leading-edge radiosurgery (LER)S is a safe and effective direct adjunctive therapy for patients with newly diagnosed GBM [104]. SRS has the benefit of reducing the toxicity and radionecrosis hazards associated with repeated wide-field fractionated radiation therapy (FRT) by conformally targeting remaining or recurrent GBM tissue [105].

## 17. Brachytherapy

Brachytherapy, which uses brain implants for radiation sources, enables very targeted radiation delivery to tiny cancers. Iodine and iridium are the most frequently used radiation sources, and they can be positioned utilizing stereotactic guidance for either permanent or transient periods [106]. Earlier detection of and differentiation between tumor recurrence and radiation necrosis is necessary. Going forward, the most effective use of brachytherapy for GBM is likely as a part of multimodal treatment with resection, chemotherapy, EBRT, and other novel therapies [106]. Glioma can be treated with brachytherapy; however, it has not been discovered to offer any benefits over traditional external beam radiation therapy (RT) [106]. By enabling the immediate intraoperative delivery of radiation therapy at the time of surgery, brachytherapy has been used to circumvent some of the drawbacks of EBRT [107]. However, using the iodine-125 (I-125) isotope in brachytherapy resulted in only modest increases in median survival with noticeably higher rates of complications such as infection, bleeding, and radiation necrosis (RN) [107]. Cesium-131 (Cs-131), a new isotope, has a number of physical and biological advantages over I-125. A study by Wernicke et al. shows that recurrent GBM patients treated with Cs-131 brachytherapy enhanced survival, exceeding 20 months. These results suggest a potential treatment for patients with recurrent GBM with a low risk of radiation necrosis development: highly conformal Cs-131 brachytherapy [108].

## 18. Proton Beam

Compared to traditional RT, photon beam therapy (PBT), which uses photon irradiation, may provide more localized radiation delivery [109], and Proton therapy (PT), or proton beam radiation therapy (PBRT), offers certain advantages to other modern photon-based conformal therapy when used in the treatment of CNS malignancies [110]. This type of RT may be used for cancers near significant brain structures because it can provide larger radiation doses to the tumor with a lower risk of damaging surrounding tissue. Like GK, PBT is administered in a single therapy session. PBT is the latest type of RT, which exerts a satisfactory curative effect, particularly in pediatric patients. In recent years, many patients have been treated with PBT worldwide. PBT can achieve a dose distribution superior to conventional external photon beam radiation. The inherent characteristics of heavy particles account for the potential advantages of PT over photon-based RT. As photons deposit energy along the X-ray beam path and the energy deposition is largest near the site of entry, conventional photon-based therapy inevitably results in an exit dosage of radiation. Protons, conversely, decelerate more quickly than photons do; as they slow down and interact with the tissue around them, they deposit more energy. This results in the Bragg peak, which deposits the highest dose at a depth specific to the tumor [111]. Compared with photon therapy, PBT is associated with obvious benefits, such as reducing the volume of irradiated normal tissue and improving the target area’s conformability and quality. However, PBT is costlier than conventional X-ray therapy. However, this increased cost may be outweighed by increasing the patient’s quality of life and reducing the expenses associated with treating late radiation-related adverse effects [112]. The available data suggest that PBT achieves good local control in some high-grade tumors with reduced toxicity but warrants the analyses of toxicity profiles for low-grade tumors. The data on the application of PT for the treatment of high-grade gliomas and GBMs, as well as ongoing clinical and translational research that may impact proton-based methods, have all been reviewed by Golf et al. [110]. There is not much evidence currently available; the sole head-to-head comparison of photon-based therapy against physical therapy for GBM is available. There is currently an inadequate amount of data to demonstrate improved results with PT in GBM.

## 19. Radiation Therapy and Its Complications

RT has been used for decades to manage benign and malignant brain tumors. Adverse side effects of RT are frequently classified as acute, early-delayed, or late-delayed, depending on when the symptoms first appear [113]. An immediate consequence that can happen hours to weeks after RT is acute encephalopathy. The symptoms worsen pre-existing neurologic deficits, including nausea, vomiting, headache, and nausea [113]. Although acute adverse effects can cause herniation and death in people with giant tumors, symptoms are typically treatable. A high dosage of radiation per fraction (>20 Gy) is the leading risk factor for the emergence of acute encephalopathy [114].

Early-delayed effects often manifest within 1–6 months after RT [115]. Lethargy, extreme weariness, and drowsiness are the hallmarks of this syndrome. Patients may experience temporary cognitive decline or worsening of localized neurologic symptoms in the first few months following radiotherapy; imaging should be used to rule out tumor growth or recurrence.

More than six months after RT, late-delayed problems emerge and are typically irreversible [116]. Radiation necrosis is possible, and edema and necrosis are two localized tissue reactions that can develop [117,118]. Risk factors include diabetes, advanced age, total radiation exposure overdoses of 55–60 Gy, and concurrent radiotherapy and chemotherapy. Radiation necrosis has a wide range of clinical symptoms; whereas some individuals are asymptomatic, others may experience convulsions or specific neurologic impairments [119]. Individuals with brain tumors are also susceptible to neuro-cognitive deficits following RT.

Further, Yamanaka et al. have systemically reviewed the characteristics and outcomes of radiation-induced gliomas. They collected information on 296 cases of radiation-induced gliomas [120]. They observed Stereotactic radiosurgery-induced gliomas in fifteen patients [120]. Twelve patients developed glioblastomas, and three patients developed anaplastic astrocytomas. Following radiosurgery treatment, the risk of radiation-induced malignancies may not be considerably different from that of conventional radiotherapy. Because secondary cancers can develop even when nearby tissues only receive low radiation doses, they should always be considered, especially in SRS cases.

## 20. Tumor-Treating Field in GBM 

The NovoTTF-100A System, approved by the FDA, is a novel therapeutic approach developed by Novocure, Ltd. in Haifa, Israel. The first-generation tumor-treating field (TTField) device (NovoTTF-100A System), which was renamed Optune later, was authorized by the FDA for recurrent GBM therapy in the year 2011 [121]. The TTField device was subsequently approved as an adjuvant therapy for freshly diagnosed GBM in 2015. This approach utilizes the targeted delivery of low-intensity, intermediate-frequency alternating electric fields to inhibit the proliferation of GBM cells. It is considered a potential treatment option for recurrent GBM patients. The delivery of TTFields to patients is facilitated by using transducers affixed to the scalp. The process of mitosis is impeded by external factors, leading to the eventual demise of the cell. Following the mapping process utilizing functional magnetic resonance imaging (fMRI), transducers are strategically positioned on the patient’s shaved scalp. Patients are recommended to employ the system for at least 18 h per day throughout each 4-week therapy cycle. Enhanced adherence to usage has been associated with improved therapeutic response and OS. The National Comprehensive Cancer Network (NCCN) incorporated the TTField device into the treatment of freshly diagnosed GBM. Despite FDA approval, uncertainty remains regarding this therapy. The evidence supporting treatment with the TTField device and its limitations have been discussed previously [121]. A randomized controlled phase III trial was conducted to evaluate the efficacy of NovoTTF-100A monotherapy in comparison to chemotherapy, as determined by the physician’s recommended treatment approach for patients with recurrent GBM. Regardless of the dearth of a significant improvement in OS, the efficacy of NovoTTF-100A was shown to be comparable to that of chemotherapy, with a success rate of 62% [122]. A study by Merkin et al. supports the use of TTFields for GBM alongside the standard-of-care treatment protocol and stipulates a practical summary, debating the present clinical and preclinical features of the treatment and their consequence on the disease course [123]. The addition of TTFields to TMZ therapy resulted in an increased OS by 4.9 months compared to patients receiving only TMZ (20.9 months and 16.0 months, respectively) [124]. In the US, Europe, and Japan, TTFields has emerged as a cutting-edge therapeutic approach that has been authorized for both freshly diagnosed and recurrent GBM. The mode of action is pertinent to other malignancies. In the continuing phase III METIS trial, Novocure is still investigating the use of TTFields in a variety of central nervous system malignancies, including brain metastases from non-small-cell lung cancer (NSCLC) [125]. TTFields are being researched in a number of additional solid tumors outside of the brain in light of the treatment’s effectiveness in GBM [125].

## 21. Limitations of TTF

Despite the advantages of TTF in the field, there are several limitations, including high price and availability issues [121].

## 22. Immunotherapy in GBM

The CNS is an immune-privileged site with limited T cell access to perform its functions. This is primarily attributed to the presence of the BBB, the absence of dedicated lymphatic channels, the low basal expression level of Major Histocompatibility Complex (MHC) class II molecules, the scarcity of antigen-presenting cells (APC), and the continuous production of immunosuppressive cytokines like tumor growth factor beta (TGF-ß). These factors collectively contribute to the significant impact on the ability of T cells to exert their immune functions [126]. Recent studies have made major contributions to our understanding of immune processes in the CNS. In a study conducted in 2015, Louveau et al. presented the notion of a typical lymphatic system located within the CNS [127]. This system was found to possess the ability to facilitate the transportation of both fluid and immune cells derived from the cerebrospinal fluid.

In recent years, there has been a substantial increase in the utilization of immunotherapy as a significant contributor to therapeutic improvements in cancer research. This is supported by several scholarly articles [128,129,130]. The advancement of immunotherapy in clinical settings has experienced notable progress due to therapeutic advancements in immune checkpoint inhibition and the utilization of chimeric antigen receptors (CAR)-modified T cells. In conjunction with the notable discoveries, the developments have significantly enhanced outcomes for cancer patients. In recent years, the FDA has witnessed a notable surge in the authorization of immunotherapy medications for cancer treatment. The pharmaceutical agents encompass monoclonal antibodies that specifically target cytotoxic-T-lymphocyte-related protein 4 (CTLA-4), programmed cell death protein 1 (PD-1), and PD-1 ligand 1 (PD-L1). In addition, CAR T cell therapy is also included in this category [128,129,130,131]. Presently, there is a lack of immunotherapies for GBM that have received approval from the FDA. However, numerous clinical trials are underway investigating the efficacy of immunotherapies in GBM patients. These trials have been motivated by the progress made in immuno-oncology for treating other types of tumors [132]. Recent research has demonstrated that the utilization of immune checkpoint inhibitors has improved OS rates among a certain group of persons diagnosed with melanoma and subsequently developing brain metastases. The implications of these findings suggest that immunotherapy could potentially serve as a promising therapeutic strategy for the treatment of CNS malignancies [133,134]. Nevertheless, the application of immunotherapy in the management of GBM remains persistently challenging as a result of the tumor-induced existence of many mechanisms of immune suppression [135].

Furthermore, it is commonly acknowledged that molecular heterogeneity inside GBM is a crucial factor in the emergence of resistance to treatment therapies. As a result, the imperative to tackle this heterogeneity has emerged as a fundamental clinical goal in the endeavor to develop efficacious immunotherapeutic approaches that are tailored especially to GBM [136]. Additionally, another study indicates that individuals with advanced solid organ malignancies who receive immunotherapy often have adverse events (AEs) that can be linked to immune-mediated mechanisms [137]. The current advancements in the field of immunotherapy pertaining to GBM present a hopeful trajectory for further investigation and improvement of GBM treatment modalities. Nevertheless, the precise clinical benefits of these advancements remain to be determined. A summary of various immunotherapies has been presented in Table 3.

## 23. Limitations in Immunotherapy

Despite encouraging outcomes, immune checkpoint inhibitors therapy ICIs are ineffective for all solid tumors, some cancers have a low response rate, and there are significant side effects [138]. An analysis of the response rate to six anti-CTLA-4 or anti-PD-1 ICIs was reported by Haslam et al. in 2019 [139]. Only about 43.6% of cancer patients were determined to be immunotherapy candidates in 2018, and the expected immunotherapy response rate was 12.46% with notable disease-specific variation. Numerous factors, such as the high degree of tumor heterogeneity and numerous immunosuppressive systems, are blamed for the poor response to immunotherapy in GBM. Although preliminary clinical trial findings were unsatisfactory, they contributed to our understanding of how GBM immunosuppression functions and newer trials are expanding on the knowledge gained from earlier trials despite the negative results [138].

## 24. CAR T Therapy in GBM

CAR T treatment offers the distinct benefit of circumventing the requirement for MHC presentation of antigens and the successive establishment of an adaptive immune response [140]. The field of hematological malignancies has provided the most compelling evidence for the clinical efficacy of CAR T-cell treatment, as demonstrated by several studies [141,142,143]. Endeavors have been commenced to extend the application of CAR T-cell treatments, which have demonstrated remarkable effectiveness in the treatment of hematological malignancies, to the treatment of solid tumors, such as GBM [144,145]. In recent studies, various clinical trials have examined the effectiveness of CAR T-cell therapy for GBM by targeting epidermal growth factor receptor variant III (EGFRvIII), interleukin (IL)13Rα2 (IL-13Ra2), and ephrin-A2 (HER2). These trials have yielded diverse outcomes, providing valuable insights [146,147]. Previous clinical trials supported the practicality of intracranial administration of IL13Rα2-specific CAR T cells for treating GBM [147]. In continuation of the preliminary findings, Brown et al. present a case study wherein a patient afflicted with recurrent multifocal GBM was administered CAR T- T-cells that specifically targeted IL-13Rα2 [148]. The study by Brown eta al; demonstrated complete regression of intracranial and spinal tumors following CAR T-cell treatment. This regression was accompanied by boosted levels of cytokines and immune cells in the cerebrospinal fluid [148]. The duration of the clinical response was observed to be 7.5 months following the commencement of CAR T-cell therapy. The precise etiology of relapse has yet to be fully understood; nevertheless, there have been documented cases of tumor recurrence accompanied by the diminished or absent expression of IL13Rα2, as reported in the following studies [148,149]. This study demonstrates that the CAR T-cells directly target tumor cells through IL13Rα2 and stimulate an innate immune response. Evidence of this includes the observed elevation of non-CAR T -T-immune cells and cytokines following each infusion. Furthermore, the treatment effectively addressed initial tumors even in cases where IL13Rα2 evasion occurred. Furthermore, a clinical trial was conducted to investigate the efficacy of HER2-CAR-modified autologous virus-specific T cells (VSTs) in patients with progressing GBM [150]. According to the study conducted by researchers, the data analysis indicated that administering autologous HER2-CAR VSTs is a safe procedure and can provide clinical advantages for individuals with progressing GBM [150].

## 25. GBM and Vaccine Therapy

The utilization of cancer vaccine therapy has demonstrated significant potential in both preventative and therapeutic contexts [151,152]. In the GBM context, cancer vaccines are specifically engineered to selectively target tumor-associated antigens to stimulate an immune response against malignant tumors. Due to the scarcity of GBM-specific antigens, the predominant targets for GBM antigens typically consist of tumor-associated antigens, restricting the inclusion of patients. A limited number of vaccination strategies have progressed to phase III clinical trials in individuals diagnosed with GBM, while several more techniques are currently in the initial phases of clinical investigation. One well-researched tumor-specific antigen is EGFRvIII, a mutant variant of EGFR that remains active and is found exclusively in 25–30% of GBM cases [153]. The utilization of genetic engineering techniques to modify T cells and enable them to express CARs that target specific antigens present in tumor cells has surfaced as an auspicious and innovative approach in cancer therapy [154]. Table 4 presents a comprehensive compilation of several vaccines. It is crucial to pick the right antigen and vaccination approach. Clinical trials for GBM vaccines are now yielding less-than-optimistic results, but with further improvement, they might become a novel therapeutic approach with enormous promise. It is necessary to increase funding for developing GBM vaccinations [155]. The target selection and vaccination approach still need to be improved. Additionally, the GBM microenvironment possesses several immune suppression mechanisms that could hinder the effectiveness of the existing vaccines. Research on adjuvant substances that might improve immunotherapy response is being conducted concurrently. Immunotherapy may result in even more long-lasting effects in GBM patients if the right target and vaccination strategy are combined with an immune modulator that either reduces the body’s ability to mount an immune response against the tumor or increases it. Several trials are now being prepared to evaluate these immunomodulators.

## 26. Nanocarrier-Mediated Therapy in GBM

A limited quantity of particles can traverse the BBB, necessitating the development of innovative technologies and delivery systems to effectively transport medications into the brain. The utilization of nanocarriers and nanotechnology in drug delivery has the potential to surmount the BBB owing to its inherent attributes such as biosafety, sustained drug release, improved solubility, enhanced drug bioactivity, BBB penetrability, and self-assembly, as supported by the scientific literature [156,157]. Nanoparticles are extensively employed to treat GBMs, and their categorization can be based on the composition of the drug carriers, such as liposomes, polymeric nanoparticles, solid lipid nanoparticles, polymeric micelles, silica, and dendrimers.

Liposomes are lipid-based vesicles that have gained significant attention in the field of drug delivery. The composition of liposomes bears resemblance to that of cellular membranes, given its construction of a hydrophilic core encased by an external phospholipid bilayer. This attribute enhances the lipophilicity of molecules and facilitates the passage of lipophilic macromolecules across the BBB. Liposomal nanoparticles have numerous advantages, such as their straightforward manufacturing, capacity to encapsulate a diverse array of anticancer medications, excellent biocompatibility, high effectiveness, lack of immunogenicity, enhanced solubility of anticancer agents, and widespread commercial availability [158]. Liposomes were first developed to encapsulate radiosensitizers and chemotherapeutic drugs, such as doxorubicin, to treat resistant tumors over twenty years ago [159]. In recent years, researchers have explored different techniques for synthesizing liposomes to treat GBMs and the use of new conjugated medicines and receptor-mediated transcytosis to enhance their transportation across the BBB [160,161,162]. An instance of improving the longevity of liposomes in circulation can be achieved through the conjugation of polyethylene glycol (PEG) to the phospholipid bilayer on the surface of the liposomes. This is attributed to the ability of PEG to facilitate the evasion of the reticuloendothelial system (RES) captured by the nanoparticles [163].

Certain receptors or antigens overexpressed on GBM cells have been identified as promising targets for advancing innovative nanotechnology in tumor treatment. For example, studies have examined the effects of IL-13-conjugated liposomes and IL-4 receptor-targeted liposomal doxorubicin in m models. These investigations have demonstrated notable reductions in tumor size when compared to liposomes that were not conjugated [164,165]. The findings suggest that receptor-conjugated liposomes do not increase toxicity in animals, highlighting its potential as a viable nanotechnology application. Moreover, the utilization of antibodies to label liposomes to target tumors can be shown [164]. Researchers have generated immunoliposomes targeting GBM cells with elevated levels of EGFR in an animal model. These immunoliposomes were found to improve the effectiveness of several anticancer medicines significantly.

Notwithstanding the widespread utilization of liposomal nanoparticle GBM therapy, several drawbacks necessitate resolution. Variations in the effects of liposomal nanoparticles are observed in different brain regions, and their ability to permeate the BBB is influenced by the specific medication or surface chemicals they carry. Recent nanocarrier and GBM therapy advancements have been summarized in Table 5 [166,167,168,169,170,171].

## 27. Magnetic Nanoparticle in GBM

The presence of chemoresistant/radioresistant cancer stem cells (CSCs) and biological barriers like the blood–brain barrier (BBB) extend hindrance to the efficacy of conventional therapies against GBM [165,172,173]. With the inception of the field of nano-theranostics, the efficacy of conventional techniques such as CHT and radiotherapy (RT) has been shown to improve significantly. This field combines therapy and diagnostics into a single nanoplatform to deliver specific and personalized therapy. In cancer, the suitability of nanoparticles (NPs) as a therapy option has been advocated for a very long time [174]. NPs generally have a varying size of 10–100 nm and can enter the tumor lumen through leaky blood vessels due to the enhanced permeability and retention (EPR) effect leading to their accumulation at the tumor site [175]. This is because the endothelium of tumor vasculature becomes more permeable than in healthy tissues, which, along with the presence of a dysfunctional lymphatic system, contributes further to the enhanced accumulation of NPs.

Targeting solid tumors using NPs for diagnostic and therapeutic purposes is a popular concept that has been explored for a very long time. The delivery of NPs to tumor regions largely relies on EPR-based passive targeting or their functionalization using suitable ligands targeting specific biomarkers (active targeting) overexpressed on the cancer cells. Over the years, several organic and inorganic NPs have been utilized for drug delivery; however, delivery efficiency could not be increased beyond 0.7% [176]. MNPs have demonstrated better effectiveness because, apart from ligand-based active targeting or EPR-mediated passive targeting, they can also be guided to the target site using an external MF. This dual targeting of MNPs to the target site is impossible with other types of NPs [173].

Over the years, NPs such as MNPs or their composites have proved their mettle as efficient drug/radiosensitizer/photosensitizer delivery platforms and in performing simultaneous therapeutic (e.g., ferroptosis therapy) and imaging functions (e.g., MRI), demonstrating incredible potential against GBM. A summary of various MNPs developed in the treatment of GBM is presented in detail in a previous review by Dhar et al. [177].

## 28. Tumor Heterogeneity in GBM

Four subgroups of GBM tumors were discovered using transcriptional profiling data from bulk tumor tissues: mesenchymal, classical, proneural, and neural. A recent study revealed 18 driver genes with varied expression profiles in several molecular subtypes, including MGMT, ATRX, H3F3A, TP53, EGFR, NES, VIM, MIK67, and OLIG2 [178]. It was discovered that the overexpression of MKi67 and OLIG2 characterized the proneural subtype, while the classical subtype had overexpression of EGFR, NES, VIM, and TP53 [178]. MGMT and VIM were overexpressed in the mesenchymal subtype, while EGFR, H3F3Q, OLIG2, S100, and TP53 were suppressed. In fact, another analysis [179] found that NES, OLIG2, VIM, and EGFR were sufficient to classify GBM into four categories. Numerous subgroups of GBM patients can be identified based on high-throughput proteomics studies that quantify the protein–protein expression patterns across a wide sample of GBM tumors. Due to particular variations in protein–protein expression networks, some patients do not fall into any categories [180,181]. Recent studies revealed that GBM biomarker expression is heterogeneous, showing heterogeneity between patients [2] and within tumor cells [2]. According to Sottoriva et al.’s investigation of samples taken from various parts of a single GBM tumor, fragments from the same tumor can be divided into several GBM subgroups through genomics. They found that one tumor clone had EGFR, CDK6, and MET amplification, while another subclone had PIK3CA amplification due to receiving a copy of chromosome 3 [182]. Tumor plasticity is caused by the occurrence of several subpopulations, which results in resistance [183] to RTK inhibitors [184] or radiotherapy [38]. Additionally, utilizing patient-derived neurosphere cultures, it was shown that the radiosensitivity of various tumor tissue locations can vary [185]. Additionally, microenvironmental stressors like hypoxia, acidosis, and reactive oxygen species may willfully lead to genomic instability, creating subpopulations resistant to de novo therapy [186]. As recently shown in other cancer types, resolving and targeting the growing cellular subpopulations in response to therapy may be a helpful strategy for preventing the emergence of drug resistance [187].

## 29. Discussion

Despite significant efforts dedicated to understanding the intricate factors contributing to the development and nature of brain tumors, the outlook for individuals diagnosed with these tumors continues to be discouraging. Heterogeneity inside and among tumors is attributed to genetic and non-genetic variables operating at various lengths and time ranges. This results in a distinctiveness observed in each tumor and patient. Therefore, adopting a comprehensive approach encompassing multiple levels of biological systems is imperative to comprehend and effectively tackle the intricacies of tumors. Adopting a personalized precision medicine strategy that considers a patient’s tumor’s distinct characteristics and biological makeup is imperative to address these obstacles. To achieve this objective, it is imperative to adopt a systems biology approach that encompasses a comprehensive understanding of the various levels of organization involved in illness causation and progression. The comprehension of illness characteristics at a systems level might aid in categorizing patients into clinically significant subtypes and provide insights into potential targets for therapeutic intervention, hence improving treatment outcomes [188].

The resistance of GBM cells and glioma-initiating cells (GICs) against current treatment techniques is facilitated by their robust DNA repair mechanisms and ability to self-renew. Therefore, developing novel treatment techniques will be necessary to manage GBM long-term effectively. The profound comprehension of the GBM microenvironment and the potential to manipulate the patient’s innate and adaptive immune system to combat the tumor constitutes the foundation of immunotherapeutic approaches that now hold promise for the battle against GBM. Although the immunotherapeutic strategy has demonstrated efficacy in various solid and hematologic neoplasms, its application in treating GBM is hindered by immune resistance and an immunosuppressive environment [189]. The future trajectory of GBM therapy will likely involve a comprehensive strategy that deviates from the current standard treatment approach of extensive tumor removal followed by chemotherapy and radiotherapy. This novel approach may entail the integration of a multifaceted immunotherapy regimen, aiming to achieve two primary objectives: the direct eradication of tumor cells and the stimulation of both innate and adaptive immune responses [190]. In a recent development, Rana et al. described a novel approach by using nanosecond pulses of 3.5 GHz radiations on the brain [191]. HPM pulses, specifically 25 pulses in U87, promoted apoptosis-related events such as ROS burst and enhanced oxidative DNA damage at higher dosages. The physiological mechanisms causing HPM-induced cell death, the safe exposure threshold for normal cells to HPM, and the therapeutic effects on cancer U87 are all better understood with the help of these discoveries. This work is timely and will help future studies as HPM technology develops. The results of this study suggest that radiation has an inhibitory effect on GBM at a particular dosage.

Nanoparticles possess distinct physical attributes, including size, shape, and surface qualities, enabling them to effectively encapsulate and transport therapeutic compounds to the brain [192]. Nanoformulations in conjunction with oral TMZ and radiotherapy have been utilized in clinical studies for patients with GBM after extensive surgical removal [192]. Furthermore, these combination treatments have demonstrated a favorable tolerance profile in most patients. Nanoformulation has shown promising results in other cancer type, such as renal cancer [193]. A similar approach would be beneficial for GBM. Using magnetic nanoparticles (MNPs) in hyperthermia treatment enables a targeted and prolonged impact on tumors. Furthermore, hyperthermia can potentially improve GBM cells’ sensitivity to radio-chemotherapy. Nonetheless, it is worth noting that none of these treatment regimens substantially enhance patients’ outcomes regarding PFS or OS as seen in previous clinical trials.

Magnetic nanoparticles (MNPs) have emerged as an auspicious nanoscale material for applying intratumoral hyperthermia therapy in GBM. Nevertheless, the replication of the effects shown in various nanoformulations on GBM cell models in actual clinical trials is hindered by the unpredictable nature of tumor heterogeneity, which is a significant barrier to the successful treatment of GBM. The available evidence on immunotherapy for GBM is generally limited, hindering its potential for major therapeutic efficacy. However, the utilization of combination therapies may offer more promising outcomes. Despite obstacles and unsatisfactory clinical outcomes in developing immunotherapy for GBM, this approach is warranted due to the therapeutic promise of this treatment and the rapidly advancing pace of research in this field.

Furthermore, the therapeutic implications of the role played by the BBB in the lack of success in treating GBM necessitate a renewed focus on enhancing technologies that disrupt the BBB, creating agents that can penetrate the BBB, and improving implanted drug delivery systems that circumvent the BBB [194]. Lastly, improved nanocarriers will play an essential role in upgrading drug delivery. Exploiting nanocarrier-based combinations would be a promising approach to enhance the therapeutic benefits of GBM. 

## 30. Conclusions

Despite the advancement in the field, GBM remains the deadliest tumor type, with restricted therapy options and poor survival. Given the genomic intricacy, cellular heterogenicity, and diverse signaling pathways, a monotherapy will unlikely be successful in GBM. An efficient treatment methodology for GBM will require combining multiple therapy approaches considering diverse oncogenic pathways. To overcome the therapy challenges, we may require an inclusive comprehension of the detailed molecular mechanisms of the therapy resistance. For instance, a recent study identified a Syx-RhoA-Dia1 signaling axis as a DNA damage regulator and therapy resistance in glioblastoma [195]. The combination of Syx depletion and TMZ synergistically inhibited cell growth in both TMZ-sensitive and TMZ-resistant cell lines. Since both TMZ and radiation therapy induce DDR, the study postulated that depletion of Syx may sensitize cells to RT [195]. Effective incorporation of different therapy tactics, including immunotherapy, nanomedicine, etc., to empower the classical treatment regimen will be the key to developing better GBM therapies. Cocktail therapy has demonstrated a growing efficacy in addressing these challenges. In a recent study, antibody cocktail-based immunotherapy that combines checkpoint blockade with dual-targeting of IL-6 and CD40 has been proposed for GBM and other solid tumors [196].

## Figures and Tables

**Table 1 brainsci-13-01536-t001:** Summary of different therapeutic approaches in GBM.

SL No	Type of Treatment	Mechanism of Action/Process
1	**Image-guided Surgery**	
	Intraoperative ultrasonography	Tumor resection
	Intraoperative MRI	
	Intraoperative fluorescence imaging	
2	**Chemotherapy**	
	Temozolomide	DNA base alkylation
	Carmustine (BCNU)	
	Lomustine (CCNU)	
	Fotemustine	
3	**Radiation based therapy (RT)**	
	2D conventional RT	DNA double-strand breaks and ROS
	3D conformal RT	
	Intensity-modulated RT	
	Stereotactic radiosurgery (SRS)	
	Brachytherapy	
	Particle RT (Proton therapy)	
4	**Inhibitor based Therapy**	
	Bevacizumab (mAb)	VEGF-A inhibition
	Irinotecan (CPT-11) (small molecule)	Inhibits topoisomerase I
	Veliparib (ABT-888) (small molecule)	PARP Inhibition
	Olaparib (AZD-2281, MK-7339) (small molecule)	
	Niraparib (MK-4827) (small molecule)	
	Pamiparib (BGB-290) (small molecule)	
	Cediranib (AZD-2171) (small molecule)	
	Gossypol (AT-101) (small molecule)	Inhibits Bcl-2, Bcl-xL and Mcl-1
	Cabozantinimb (XL-184) (small molecule)	Tyrosine kinase inhibitor
	Erlotinib	EGFR inhibitor
	Gefitinib	
	Depatuxizumab mafodotin (ABT-414)	EGFR and tubulin
	Imatinib	Tyrosine kinase inhibitor
	Dasatinib	
	Sorafenib	
	Sunitinib	
	Temsirolimus (CCI-779)	mTOR inhibitor
	Everolimus	
5	**Nanoformulation(Liposomes)**	
	2B3–101 PEGylated liposomes	Target GSH/GSH transporters
	SGT-53 Cationic liposomes	Target Scfv/TfR
	Liposomal irinotecan	Convection enhanced delivery (CED)
6	**Immunotherapy**	
	Cemiplimab	Checkpoint inhibitor that binds to PD-1
	Nivolumab	
	Rindopepimut peptide vaccine	Targets EGFR deletion mutation EGFRvIII
	DCVax^®^-L	DCs are primed to recognize tumor-specific antigens
	VB-111 (Ofranergene obadenovec) gene therapy using an adenovirus type 5 vector	Virus carries a trans-gene for chimeric death receptor that connects Fas to hTNF receptor 1.
	CAR T cell therapy	Engineered T cells are that express receptors against specific tumor markers.
7	**Other approach**	
	Laser interstitial thermal therapy (LITT)	Thermal ablation of tumor tissue
	Tumor Treating Fields (TTF)	Disrupts mitotic cell division

**Table 2 brainsci-13-01536-t002:** Summary of selected clinical trials of drug combinations for the treatment of GBM.

SL No	Drugs Combination	Mechanism of Action	Class	Phase	Clinical Trial ID	Status
1	Bevacizumab; Irinotecan	Anti-VEGF antibody; Topoisomerase I inhibitor	Recurrent Gliomas	Phase II	NCT00921167	Completed
2	O6-Benzylguanine; Temozolomide	O6-alkylguanine-DNA alkyltransferase inhibitor; Alkylating agent	Temozolomide- resistant malignant glioma		NCT00613093	
3	Imatinib; Hydroxyurea	Tyrosine kinase inhibitor; ribonucleoside diphosphate reductase inhibitor	Recurrent/ progressive grade II low-grade Glioma		NCT00615927	
4	Cediranib; Lomustine	Tyrosine kinase; Alkylating agent	Recurrent GBM	Phase III	NCT00777153	
5	Sorafenib; Temsirolimus	Tyrosine kinase inhibitor; mTOR inhibitor		Phase I/II	NCT00329719	
6	Bevacizumab; Sorafenib	Anti-VEGF antibody; Tyrosine protein kinases		Phase II	NCT00621686	
7	Bevacizumab; Temsirolimus	Anti-VEGF antibody; mTOR inhibitor			NCT00800917	
8	Erlotinib; Sirolimus	Tyrosine kinase inhibitor; mTOR inhibitor			NCT00672243	
9	Vorinostat; Bortezomib	Deacetylase inhibitor; Proteasome inhibitor			NCT00641706	
10	Bevacizumab; Erlotinib	Anti-VEGF antibody; Tyrosine kinase inhibitor;			NCT00671970	
11	Temozolomide; SGT-53	Alkylating agent; Liposome-p53 DNA			NCT02340156	Ongoing
12	Glasdegib; Temozolomide	Inhibits SHH pathway interfering with cancer stem cells and endothelial migration; Alkylating agent	Newly diagnosed GBM	Phase IB/II	NCT03466450	
13	Bevacizumab; Capecitabine	Anti-VEGF antibody; Target myeloid-derived suppressor cells	Recurrent GBM	Phase I	NCT02	

**Table 3 brainsci-13-01536-t003:** Summary of immunotherapy and their status in the treatment of GBM.

Type	Name	Drug	Combinations	Status	Clinical Trials ID
**1**	Dual Checkpoint Blocker	CTLA-4 (ipilimumab)	PD-1 (nivolumab) therapy	Phase I	NCT02311920
		PD-1 (nivolumab)	Anti-LAG-3 (BMS 986016)/anti CD137 (urelumab)		NCT02658981
			anti-CD-27 (varlilumab)	Phase I/II	NCT02335918
			Intratumoral IDO1 inhibitor (INT230-6)		NCT03058289
			IDO1 inhibitor (epacadostat)		NCT02327078
		PD-L1 (durvalumab)	CTLA-4 (tremelimumab)	Phase II	NCT02794883
**2**	Vaccines	PD-1 (pembrolizumab)	HSPPC-96		NCT03018288
					AVeRT
		PD-1 (nivolumab)	pp65 DC	Phase I	NCT02529072
			DCVAX-L		NCT03014804
**3**	Virus	PD-1 (pembrolizumab)	DNX-2401		NCT02798406
**4**	Radiation Therapy	Pembro	Hypofractionated stereotactic irradiation		NCT02313272
		Nivo	SRS + Valproic acid		NCT02648633
			hypofractionated stereotactic irradiation		NCT02829931
		PD-L1 (durvalumab)	Hypofractionated stereotactic irradiation	Phase I/II	NCT02866747
**5**	Laser Treatment	MK-3475	MRI-guided laser ablation		NCT02311582
**6**	CSF-1R inibitor	Nivo	CSF-1r inhibitor (BLZ945)		NCT02526017
		PD-1 (PDR001)	CSF-1r inhibitor (FPA008)	Phase I	NCT02829723

**Table 4 brainsci-13-01536-t004:** Vaccines in GBM therapy.

SL No	Name	Clinical Trial ID	Class	Status
1	Rindopepimut	NCT01480479	Newly diagnosed GBM (nGBM)	Phase III
		NCT00458601		Phase II
		NCT01498328	Recurrent GBM (rGBM)	
2	ADU-623	NCT01967758	rAA, rGBM	Phase I
3	HSPPC-96	NCT00905060	Newly diagnosed GBM	Phase II
4	HSPPC-96	NCT02122822		Phase I
		NCT02722512	Newly diagnosed or recurrent pediatric HGG, ependymoma	Phase I
		NCT01814813	Recurrent GBM	Phase II
		NCT03018288	Newly diagnosed GBM	
5	IDH1 R132H Derivative	NCT02454634	nAA, nAO, nGBM	Phase I
6	K27M peptide	NCT02960230	Newly diagnosed GBM	
7	SurVaxM	NCT02455557		Phase II
8	DCs vaccine (DCVax)	NCT00045968		Phase III
9	DCs vaccine (DCVax)	NCT02146066	nGBM, rGBM	Expanded access
10	DCs vaccine (brain tumor stem cells mRNA loaded)	NCT00890032	Recurrent GBM	Phase I
11	DCs vaccine (brain tumor stem cells as antigen)	NCT01171469	rAA, rGBM, recurrent medulloblastoma, recurrent ependymoma	
12	DCs vaccine (tumor stem cell-loaded)	NCT02820584	Recurrent GBM	
13	DCs vaccine (fusion peptide loaded)	NCT01522820		
14	DCs vaccine (tumor mRNA loaded)	NCT02709616	Newly diagnosed GBM	Phase I/II
15	DCs vaccine (pp65 RNA loaded)	NCT00639639		Phase I
16	DCs vaccine (pp65 RNA loaded)	NCT02465268		Phase II
		NCT02366728		
17	DCs vaccine (autogenic glioma stem-like cells (A2B5+) loaded)	NCT01567202	Newly diagnosed recurrent GBM	
18	DCs vaccine (tumor lysate loaded)	NCT01204684	nAA, rAA, nAO, rAO, nGBM, rGBM	
19	DCs vaccine (RNA loaded)	NCT00626483	Newly diagnosed GBM	Phase I
20	DCs vaccine (tumor lysate loaded)	NCT01957956		
21	DCs vaccine (peptide loaded)	NCT02049489	Recurrent GBM	
22	DCs vaccine (Wilms’ tumor 1 mRNA loaded)	NCT02649582	Newly diagnosed GBM	Phase I/II
23	DCs vaccine + tumor lysate boost	NCT01808820	rAA, rGBM	Phase I
24	DCs vaccine (allogenic GBM stem-like cell lysate loaded)	NCT02010606	Newly diagnosed and recurrent GBM	
25	DCs vaccine (tumor lysate loaded)+ nivolumab	NCT03014804	Recurrent GBM	Phase II
26	Vaccine derived from tumor lysate	NCT01400672		Phase I
27	HSCs, DCs vaccine, CTLs	NCT01759810		Phase II/III
28	Bevacizumab and TAA, Poly-ICLC, KLH	NCT02754362		Phase II
29	SL-701, poly-ICLC, bevacizumab	NCT02078648		Phase I/II
30	ICT-107	NCT02546102	Newly diagnosed GBM	Phase III
31	IMA950, Poly ICLC	NCT01920191		Phase I/II
32	IMA950, GM-CSF	NCT01222221		Phase I
33	Personalized peptide vaccine, Poly ICLC	NCT02510950		Phase 0

**Table 5 brainsci-13-01536-t005:** Summary of clinical trials using nanotechnology- and nanocarrier-based delivery systems for treating glioblastoma multiforme.

SL No	Name	Composition	Status	References
1	Nanothermotherapy	Nanoparticles (Thermotherapy and Magnetic iron-oxid) and A radiotherapy (Low dose)	Phase II	[166]
2	EDV-doxorubicin	Combination of EnGenelC delivery vehicle (EDV)-doxorubicin and radiation and oral TMZ	Phase I	[167]
3	Interleukin-12	IL-12 gene in semliki Forest virus vector capsulated in cationic liposomes	Phase I, II	[168]
4	5-fluorouracil	5-fluorouracil-releasing microspheres and radiotherapy	Phase II	[169]
5	Caelyx, PEG-Dox	Combination of Pegylated liposomal doxorubicin, TMZ and radiotherapy	Phase I, II	[170]
6	PEG-Dox	Radiotherapy and surgery followed by TMZ and Pegylated liposomal doxorubicin	Phase II	[171]

## Data Availability

Not applicable.

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
