# Peer review of "Advances in Glioblastoma Therapy: An Update on Current Approaches"

_brainsci, 2023, doi:10.3390/brainsci13111536_

Round 1

Reviewer 1 Report

Comments and Suggestions for Authors

Review on Advances in Brain Tumors Therapy: An Update on Glioblastoma Multiform

I have completed my review of manuscript Brainsci-2649363, entitled, Advances in Brain Tumors Therapy: An Update on Glioblastoma Multiform.”

This review article explores the formidable challenges associated with treating glioblastoma multiforme (GBM), a highly malignant brain tumor with a grim prognosis. The paper highlights the inadequacies of current GBM therapies due to factors like tumor heterogeneity, the blood-brain barrier, drug resistance mechanisms, and glioma stem cells. To address these complexities, the paper advocates for a multifaceted approach, emphasizing combination therapies that target various oncogenic pathways. It discusses promising strategies, including immunotherapy, nanomedicine, and innovative agents. Furthermore, the paper underscores the necessity of a comprehensive understanding of therapy resistance mechanisms. It recognizes that the future of GBM treatment lies in combining diverse therapeutic tactics and leveraging emerging technologies. In conclusion, GBM remains a formidable clinical challenge, and a successful treatment methodology will likely require a holistic approach that integrates multiple therapeutic strategies to overcome its multifaceted nature.

The subject of this review is very important. Unfortunately, the quality of the paper is not well enough to be positive with the present form of the manuscript. Major revision is required by addressing each comment given below.

Comments for authors

Comment 1: In evaluating this manuscript as a review article, it becomes evident that the introduction section lacks the necessary depth to fully convey the field's significance. A solid foundation for the review article relies on comprehensive background information supported by recent literature.

Comment 2: The author should explicitly highlight the novelty of this review in the abstract compared with the existing literature

Comment 3: The title "Advances in Brain Tumors Therapy" is quite broad, and it's crucial to specify the specific time frame and focus of this review. The title of the manuscript should be appropriately aligned with the content of the literature review.

Comment 4: The paper discusses the challenges in treating glioblastoma multiforme (GBM). Can you provide a critical analysis (based on available literature) of the factors contributing to the ineffectiveness of current GBM treatments, and how to address these challenges through combination therapies?

Comment 5: The paper mentions the role of tumor heterogeneity in GBM. Could you elaborate on the molecular and clinical implications of tumor heterogeneity in GBM, and how this might influence the choice of therapeutic strategies? This will be the key for this review article.

Comment 6: In the abstract, the authors mentioned the need for "pharmacological cocktail therapy" to address the challenges of GBM treatment. Can you provide specific examples from the literature of successful pharmacological cocktail approaches in GBM, and what are the key considerations in designing such combinations? This information will be of paramount importance for the readers of this review.

Comment 7: The conclusion emphasizes the importance of a comprehensive understanding of therapy resistance mechanisms. Could you elaborate on any recent breakthroughs or novel insights regarding therapy resistance in GBM that have been discussed in the paper? In my opinion, the conclusion needs be to more informative.

Comment 8: Authors are encouraged to incorporate a discussion of a very recent study that demonstrated the therapeutic potential of pulsed high-power microwave treatment at specific doses for glioblastoma within the context of this review.

Article information: ROS production in response to high-power microwave pulses induces p53 activation and DNA damage in brain cells: Radiosensitivity and biological dosimetry evaluation, Front. Cell Dev. Biol. 11 (2023). https://doi.org/10.3389/fcell.2023.1067861.

Comment 9: Several sections require expansion and the inclusion of references to support their claims. Most of the sections are too brief and may not adequately convey their key points. Also, only 1 or 2 references are provided to cover the whole section, which is unacceptable. It is crucial to broaden the literature review beyond just one or two sources for each section for a full review article.

Comment 10: The paper contains errors and typos that make it difficult to understand and distort its intended meaning. I encourage authors to reread carefully and fix any grammatical errors.

I hope the comment will help the authors to improve the manuscript. I would like to see this manuscript after the suggested revisions.

Comments on the Quality of English Language

The paper contains some errors and typos that make it difficult to understand and distort its intended meaning. I encourage authors to reread the manuscript carefully and fix any grammatical errors.

Author Response

Response to Reviewer 1
Comment 1: In evaluating this manuscript as a review article, it becomes evident that the introduction section lacks the necessary depth to fully convey the field's significance. A solid foundation for the review article relies on comprehensive background information supported by recent literature.
Response: We thank the reviewer for this valuable suggestion. We have highlighted the significance in the introduction of this revised manuscript. We believe these highlights will provide more depth to convey the field's significance. Please see the changes highlighted.
Comment 2: The author should explicitly highlight the novelty of this review in the abstract compared with the existing literature.
Response: We thank the reviewer for this suggestion. We have now highlighted the significance in this revised manuscript.
Comment 3: The title "Advances in Brain Tumors Therapy" is quite broad, and it's crucial to specify the specific time frame and focus of this review. The title of the manuscript should be appropriately aligned with the content of the literature review.
Response: We have revised the title with the present one, "Advances in Glioblastoma Therapy: An Update on Current Approaches. We hope this revised title is appropriate for the review.
Comment 4: The paper discusses the challenges in treating glioblastoma multiforme (GBM). Can you provide a critical analysis (based on available literature) of the factors contributing to the ineffectiveness of current GBM treatments, and how to address these challenges through combination therapies?
Response: We have now provided a detailed analysis on the suggested topic "factors contributing to the ineffectiveness of current GBM treatments, and how to address these challenges through combination therapies" under the Therapeutic resistance in GBM in the main article. Please see Lines 123-177.
Comment 5: The paper mentions the role of tumor heterogeneity in GBM. Could you elaborate on the molecular and clinical implications of tumor heterogeneity in GBM, and how this might influence the choice of therapeutic strategies? This will be the key for this review article.
Response: Thank you for this suggestion. We have included a section on tumor heterogeneity and elaborated its implication in GBM therapy. Please see 773-799 in the text.
Comment 6: In the abstract, the authors mentioned the need for "pharmacological cocktail therapy" to address the challenges of GBM treatment. Can you provide specific examples from the literature of successful pharmacological cocktail approaches in GBM, and what are the key considerations in designing such combinations? This information will be of paramount importance for the readers of this review.
Response: We have provided recent literature describing specific examples of cocktail therapy in GBM. Please see the highlights in Lines 882-885.
Comment 7: The conclusion emphasizes the importance of a comprehensive understanding of therapy resistance mechanisms. Could you elaborate on any recent breakthroughs or novel insights regarding therapy resistance in GBM that have been discussed in the paper? In my opinion, the conclusion needs be to more informative.
Response: We thank the reviewer for this suggestion. We have updated the conclusion to reflect the novel insight regarding therapy resistance. Please see the new changes.
Comment 8: Authors are encouraged to incorporate a discussion of a very recent study that demonstrated the therapeutic potential of pulsed high-power microwave treatment at specific doses for glioblastoma within the context of this review.
Article information: ROS production in response to high-power microwave pulses induces p53 activation and DNA damage in brain cells: Radiosensitivity and biological dosimetry evaluation, Front. Cell Dev. Biol. 11 (2023). https://doi.org/10.3389/fcell.2023.1067861.
Response: We thank the reviewer for this suggestion. We have included this article in the discussion. Please see Line 830-838.
Comment 9: Several sections require expansion and the inclusion of references to support their claims. Most sections are too brief and may not adequately convey their key points. Also, only 1 or 2 references are provided to cover the whole section, which is unacceptable. It is crucial to broaden the literature review beyond just one or two sources for each section for a full review article.
Response: We appreciate the reviewer's comments. We have expanded these sections below and included more references. Please see the highlighted sections in the revised manuscript. We believe that these changes will address the reviewer's comments.
Comment 10: The paper contains errors and typos that make it difficult to understand and distort its intended meaning. I encourage authors to reread carefully and fix any grammatical errors.
Response: Thank you for these suggestions. We have carefully checked the entire manuscript and addressed these errors and typos throughout the manuscript.

Reviewer 2 Report

Comments and Suggestions for Authors

With ongoing research and clinical trials, a periodical resume of the current knowledge, even in such extensively described tumor as GBM, is of value. Therefore the ms subject meets the needs of the scientific community.
The ms presents a great amount of information. However, it lacks consistency and leading thought. It is visible, that the authors do not have a lot of experience with the neurooncological field in clinical practice.

The title "Advances in Brain Tumors Therapy: An Update on Glioblastoma Multiform" should be reformulated. It is not clear whether it will be about GBM or brain tumors in general. I assume GBM. The BTs as a whole group are very diverse and so is treatment strategy - incomparable.

The nomenclature has to be corrected ("glioblastoma multiform", "grade IV"). I recommend the authors acknowledge the up-to-date CNS tumor classification (https://www.ncbi.nlm.nih.gov/pmc/articles/PMC8328013/).

line 34-35: logically and practically it is the maximal possibly safe tumor resection followed by radiotherapy and chemotherapy with temozolomide.

In general the "Introduction" is too long, and totally mixed up - it starts with TTF, then goes to standard technique 5-ALA, switches to metastases, then goes to "CNS cancer" (very general term, while the ms is about glioblastoma), glioma treatment, QoL. I would recommend rearrangement and decision about logical argumentation - probably after mentioning the malignancy and survival, other epidemiological issues (up-to-date literature) highlight the reasons for malignancy, non-total resection, resistance to standard treatments, subgroups of glioblastoma with better survival and reasons for that survival, ...
However the authors want it, but it has to be consistent and considering GBM.
I recommend reading: https://pubmed.ncbi.nlm.nih.gov/32328653/.

The tables should be better visually edited.

I didn't notice mentioning the role of focused ultrasound in GBM treatment.

"Surgical method of removing the tumor" - too long chapter mixing the description of any brain surgery with keywords of neurophysiological monitoring. Completely irrelevant to the subject. The ms title is "Advances in Brain Tumors Therapy: An Update on Glioblastoma Multiform". The surgical principles are maximal possible and neurologically safe - important aspects are resection of CE zone, supramarginal resection, PFS/OS depending on EOR, continous monitoring with tractography monitoring during resection, visualization (5-ALA, ICG, SF) - gain of EOR, EOR monitoring - US/MRI.

Further - another mentioning of metastases!

Besides exclusive single cases, endoscopic treatment has nothing to do with GBM.

Tubular retractors go to safe neurological resection protecting the fibres.

In the chemotherapy section - maybe a table summarizing the efficacy of the drugs, and stating whether there are firs/second line drugs, as well as relation to molecular markers, should be included.

Radiotherapy - WBRT and SRS are first line brain metastases approaches. Gamma knife/Cyber Knife/linear accelerator are just different mechines for SRS.
Radiotherapy has summarizing paragraph about complications - other methods do not.

Concluding - the authors should decide what is the subject of the ms and organize logiaclly their argumentation and treatment methods around that subject.

Comments on the Quality of English Language

There are small grammatical/spelling/punctuation/styllistic mistakes. Needs one thorough reading.

Author Response

Response to Reviewer 2
With ongoing research and clinical trials, a periodical resume of the current knowledge, even in such extensively described tumor as GBM, is of value. Therefore the ms subject meets the needs of the scientific community. The ms presents a great amount of information. However, it lacks consistency and leading thought. It is visible, that the authors do not have a lot of experience with the neurooncological field in clinical practice. 1. The title "Advances in Brain Tumors Therapy: An Update on Glioblastoma Multiform" should be reformulated. It is not clear whether it will be about GBM or brain tumors in general. I assume GBM. The BTs as a whole group are very diverse and so is treatment strategy - incomparable.
Response: We appreciate the reviewer for their critical review and valuable suggestions. We have formulated the Title as "Advances in Glioblastoma Therapy: An Update on Current Approaches". We hope that the new title matches the theme of the review article.
2. The nomenclature has to be corrected ("glioblastoma multiform", "grade IV"). I recommend the authors acknowledge the up-to-date CNS tumor classification (https://www.ncbi.nlm.nih.gov/pmc/articles/PMC8328013/).
Response: Thank You. We have made the changes as per the suggestion.
In general, the "Introduction" is too long, and totally mixed up - it starts with TTF, then goes to standard technique 5-ALA, switches to metastases, then goes to "CNS cancer" (very general term, while the ms is about glioblastoma), glioma treatment, QoL. I would recommend rearrangement and decision about logical argumentation - probably after mentioning the malignancy and survival, other epidemiological issues (up-to-date literature) highlight the reasons for malignancy, non-total resection, resistance to standard treatments, subgroups of glioblastoma with better survival and reasons for that survival, ... However, the authors want it, but it has to be consistent and considering GBM. I recommend reading: https://pubmed.ncbi.nlm.nih.gov/32328653/.
Response: We thank the reviewer for this suggestion. We have rearranged the introduction as suggested. We hope the introduction has improved after the re-arrangement. Please see the highlighted changes in the introduction.
The tables should be better visually edited.
Response: We apologize for the poor quality of the table. We have now replaced them with a new table of better resolutions.
"Surgical method of removing the tumor" - too long chapter mixing the description of any brain surgery with keywords of neurophysiological monitoring. Completely irrelevant to the subject. The ms title is "Advances in Brain Tumors Therapy: An Update on Glioblastoma Multiform". The surgical principles are maximal possible and neurologically safe - important aspects are resection of CE zone, supramarginal resection, PFS/OS depending on EOR, continous monitoring with tractography monitoring during resection, visualization (5-ALA, ICG, SF) - gain of EOR, EOR monitoring - US/MRI.
Response: We thank the reviewer for these comments. We have updated the Surgery section to make it precise.
Further - another mentioning of metastases!
Response: Thank you for this suggestion.
Besides exclusive single cases, endoscopic treatment has nothing to do with GBM.
Response: Thank you for this comment. To be more specific to the theme of this review, we have removed this sections.
Tubular retractors go to safe neurological resection protecting the fibres.
Response: We have removed this portion as suggested.
In the chemotherapy section - maybe a table summarizing the efficacy of the drugs, and stating whether there are firs/second line drugs, as well as relation to molecular markers, should be included.
Response: We thank the reviewer for this suggestion. We have attempted to discuss the chemotherapeutic drugs in GBM in Tables 1 and 2. We apologies that we didn’t address this comment fully, but we hope the list of drugs provided in these Tables will be informative as suggested by the reviewers in their comments.
Radiotherapy - WBRT and SRS are first line brain metastases approaches. Gamma knife/Cyber Knife/linear accelerator are just different machines for SRS. Radiotherapy has summarizing paragraph about complications - other methods do not.
Response: Thank you for these suggestions and comments. We have corrected the issues. We have also included the complication in other methods. Please see the highlighted changes. We hope the point-by-point response to each comment has addressed the reviewer's concerns and improved the manuscripts satisfactorily.

Reviewer 3 Report

Comments and Suggestions for Authors

The authors present a complete and well-written review article discussing in depth the advances in brain tumor therapy. I recommend the authors add these additional references and other minor revisions. Please, also change ‘temozolomide’ to ‘TMZ’ throughout the text. Congratulations, this is a very well presented manuscript about the topic.

-The authors should write a short summary chapter mentioning the known mechanisms associated to chemoresistance and radiation resistance, as there are the standard care of treatment. Do not focus on details; it could be 2-3 paragraphs highlighting these mechanisms and it could be after the introduction. There are some of the mechanisms that should be mentioned: presence of transport proteins making the passage of substances difficult through the BBB (DOI: 10.1016/j.drup.2015.02.002), cell factors present in the tumor microenvironment like astrocytes and macroglia (10.1186/s12967-014-0278-y; DOI: 10.1038/nn.4185), non-cell factors in the tumor microenvironment such as pH, hypoxia and the extracellular matrix, several genetic alterations such as EGFR amplification, loss of PTEN, mutation in P53, etc, higher repair activity of the DNA which hampers the DNA breaks induced by TMZ and/or radiation (DOI: 10.1007/s12035-022-02915-2), among others.

-Some sentences in the Table do not start with a Capital letter, such as ‘convection enhanced delivery (CED)’, ‘disrupts mitotic cell division’, etc

-The authors should describe these two important studies in the chapter ‘Radiation therapy and its complications’: (1) it has been revealed that the application of radiation in transgenic mice with deletions tumor suppressors that are frequently absent in GBM is more effective in inducing tumors de novo compared to X-rays (DOI: 10.1038/onc.2014.29), and (2) these radiation-initiated gliomas exhibit histologic grade IV attributes. Therefore, in the presence of tumor suppressor loss, radiation could contribute to the formation of high-grade gliomas (DOI: 10.1158/0008-5472.CAN-19-0680).

Comments on the Quality of English Language

English fine is adequate. I suggest to use the free software Grammarly in case there is a need to polish the manuscript a bit more.

Author Response

Response to Reviewer 3
The authors present a complete and well-written review article discussing in depth the advances in brain tumor therapy. I recommend the authors add these additional references and other minor revisions. Please, also change 'temozolomide' to 'TMZ' throughout the text. Congratulations, this is a very well presented manuscript about the topic.
-The authors should write a short summary chapter mentioning the known mechanisms associated to chemoresistance and radiation resistance, as there are the standard care of treatment. Do not focus on details; it could be 2-3 paragraphs highlighting these mechanisms and it could be after the introduction. There are some of the mechanisms that should be mentioned: presence of transport proteins making the passage of substances difficult through the BBB (DOI: 10.1016/j.drup.2015.02.002), cell factors present in the tumor microenvironment like astrocytes and macroglia (10.1186/s12967-014-0278-y; DOI: 10.1038/nn.4185), non-cell factors in the tumor microenvironment such as pH, hypoxia and the extracellular matrix, several genetic alterations such as EGFR amplification, loss of PTEN, mutation in P53, etc, higher repair activity of the DNA which hampers the DNA breaks induced by TMZ and/or radiation, among others.
-Some sentences in the Table do not start with a Capital letter, such as 'convection enhanced delivery (CED)', 'disrupts mitotic cell division', etc
-The authors should describe these two important studies in the chapter' Radiation therapy and its complications': (1) it has been revealed that the application of radiation in transgenic mice with deletions tumor suppressors that are frequently absent in GBM is more effective in inducing tumors de novo compared to X-rays, and (2) these radiation-initiated gliomas exhibit histologic grade IV attributes. Therefore, in the presence of tumor suppressor loss, radiation could contribute to the formation of high-grade gliomas.
Response: We thank the reviewer for these valuable comments and suggestions. We have made the suggested changes. Please see the highlighted regions in the text Line 122 – 177, and Line 519-526.
We have also corrected the Tables as suggested. We hope that the revised manuscript has addressed the reviewer's concerns and the manuscript has improved significantly.

Round 2

Reviewer 1 Report

Comments and Suggestions for Authors

The authors have taken great care to address every one of my comments and concerns in their revised manuscript, resulting in a notable enhancement in its quality. I recommend the publication of the paper in its present form.